# The Mineral Composition of Date Palm Fruits (*Phoenix dactylifera* L.) under Low to High Salinity Irrigation

**DOI:** 10.3390/molecules26237361

**Published:** 2021-12-04

**Authors:** Rania Dghaim, Zied Hammami, Rola Al Ghali, Linda Smail, Dalia Haroun

**Affiliations:** 1Department of Health Sciences, College of Natural and Health Sciences, Zayed University, Dubai P.O. Box 19282, United Arab Emirates; rolaghali@gmail.com (R.A.G.); Dalia.Haroun@zu.ac.ae (D.H.); 2Crop Diversification and Genetics Section, International Center for Biosaline Agriculture (ICBA), Dubai P.O. Box 14660, United Arab Emirates; z.Hammami@biosaline.org.ae; 3Department of Mathematics and Statistics, College of Natural and Health Sciences, Zayed University, Dubai P.O. Box 19282, United Arab Emirates; Linda.Smail@zu.ac.ae

**Keywords:** date palm, salinity, mineral, percent Daily Value (% DV), United Arab Emirates (UAE)

## Abstract

Adaptability to salinity varies between different varieties of date palm trees. This research aims to explore the long-term impact of different salinity irrigation levels on the mineral content of 13 date palm varieties grown in the United Arab Emirates (UAE). Date varieties were grown using three irrigation water salinity levels of 5, 10 and 15 dS m^−1^. The mineral composition (B, Ca, Cu, Fe, K, Mg, Na, P and Zn) of date palm fruits was determined using Inductively Coupled Plasma-Optical Emission Spectrometry (ICP-OES). High salinity levels showed no effect on the mineral content of Ajwat AlMadinah, Naghal, Barhi, Shagri, Abu Maan, Jabri, Sukkari and Rothan varieties. All date varieties remained good sources of dietary potassium, magnesium, manganese and boron even at high salinity levels. Increased salinity had no effect on the percent Daily Value (%DV) categories of most of the analyzed minerals. While no genotypes showed a general adaptation to different saline environments, Barhi, Ajwat Al Madinah, Khinizi, Maktoumi and Shagri varieties were more stable towards salinity variation. In the UAE, the genotype x saline-environment interaction was found to be high which makes it impossible to attribute the variation in mineral content to a single varietal or salinity effect.

## 1. Introduction

The date palm (*Phoenix dactylifera* L.) is traditionally cultivated in arid regions of the world, including the Arabian Peninsula. It is one of the oldest fruit trees, a key component of the food system, and is recognized as a symbol of prosperity in the Arab world. Accordingly, the date palm is appreciated for its high nutritive, economic as well as social values. The production, use, and processing of dates are continually increasing in all parts of the world. There are over 1500 known date palm varieties, and nearly 250 of those are produced in the Arabian Peninsula. The United Arab Emirates (UAE) has the largest number of date palms of any single country in the world. It has over 40 million date palm trees, with more than 200 cultivars, 68 of which have commercial importance. The UAE ranks among the top five major date producing countries in the world [1]. The export of dates from the UAE exceeded 275,862.901 tons in 2016 [2]. The UAE is also among the countries with the highest consumption of dates. Tamar and Rutab are the most consumed dates in the UAE. The average daily consumption per capita ranges between 8 and 10 dates (72–114.3 g) [3,4].

The physical scarcity of water and salinity represent a serious concern for food production in the Middle East and North African (MENA) region. The date palm is known to tolerate several biotic and abiotic stresses and is known to be the most salt-tolerant of all halophyte crops. The palm tree has a minimal water demand, and tolerates harsh weather and high salinity [5,6]. Nevertheless, due to the large number of date palm trees grown in the UAE, a large amount of water is used for irrigation. For example, the irrigation of date palms currently accounts for about one-third of all groundwater used in the UAE [7]. Moreover, the salinization of both surface and groundwater systems has been exacerbated by high evapotranspiration rates. The salinity is further exacerbated by the noticeable effects of climate change on increasing temperatures and declining rainfall [8]. Date palm growth and production are adversely affected by increasing soil and water salinities.

Soil salinity poses a serious threat to agricultural productivity and food security worldwide. More than 6% of the total land area is affected by salt, which pertains to more than 800 million hectares of arable land [9]. Soil salinity is more pronounced in arid and semi-arid lands, which face other agricultural impediments such as water shortage and land degradation [10]. This is particularly true for the UAE. The UAE is facing multiple challenges in managing water resources. These include the scarcity of freshwater resources, a saltwater intrusion of aquifers, and overexploitation of groundwater resources. The concern over water scarcity and its impact on the environment and agriculture has prompted researchers to explore other water source alternatives, including saline (brackish) water for irrigation. Therefore, to exploit saline water and/or salt-affected land, it is critical to identify appropriate crops of plant species and varieties that have a good range of salt tolerance. Plants that adapt to saline soils and attain normal growth and development are known as halophytes [11].

Adaptability to salinity in plants is a complex process that varies among plant species, cultivars of the same species, and even among individuals of the same cultivars [12]. The physiological basis of this tolerance and sensitivity is not fully known. In general, two types of adaptation mechanism to soil salinity are proposed: (1) dilution or exclusion and extrusion, and (2) osmoregulation [11].

Salt stress significantly affects and limits crop production and growth. In low–moderate salinity conditions, plants metabolize normally with no symptoms of injury. However, they need more energy to maintain a normal metabolism, causing a reduction in growth and yield. The effect on growth is attributed to osmotic effects, ion toxicity, nutrient uptake imbalance, or combinations of these factors. Additionally, high salinity can cause significant morphological changes in the plant response, such as in the plant height, leaf production, and collar girth of different varieties [5,13,14,15]. Date cultivars are classified into two distinct groups based on their growth response to salinity: a salt-sensitive group with a significant reduction in shoot growth, and a salt-tolerant group [16].

In 2001–2002, a long-term experiment was launched by the International Center for Biosaline Agriculture (ICBA), Dubai, UAE in collaboration with the UAE Ministry of Environment and Water, to evaluate the salt tolerance of elite date varieties that are common to the UAE and the gulf region. Salt tolerance studies on the date palm have focused on the effect of salinity on growth and yield, with little or no data available on the mineral quality of the date fruits irrigated with highly saline water. Generally, information on the salt tolerance of date palm varieties and assessments of the impact of long-term use of marginal quality irrigation on fruit quality are scarce and limited. A large gap in understanding the impact of salinity on date palms is therefore evident. The main objectives of this research are to explore the effect of high salinity irrigation on the mineral content of the fruit of elite date varieties commonly grown in the UAE, and to identify the salt-tolerant varieties which provide a significant contribution to the percent Daily Values (%DV) of minerals.

## 2. Results

### 2.1. General Mineral Profile

The nutritional quality of date palms is, in part, associated with their major constituents, including minerals. Dates contain at least 15 essential minerals including phosphorus, potassium, sodium, zinc, manganese, magnesium, copper, and iron [17]. Minerals are essential supplements for bones, teeth, soft tissues, hemoglobin, muscles and nerve cells [18]. Mineral content varies depending on the cultivar, ripening stage, agronomical practices and environmental conditions [17,19]. The mineral composition of thirteen varieties of date palm fruits is averaged at three salinity levels, and mean values ± SD are summarized in Table 1.

The tested varieties contained considerable amounts of minerals (Table 1). In particular, potassium was the highest with concentrations in the range 6306.95–8293.69 mg/kg, followed by phosphorus (611.60–852.03 mg/kg), calcium (571.95–766.00 mg/kg), magnesium (496.55–717.26 mg/kg), sodium (207.29–429.98 mg/kg), boron (6.32–12.84 mg/kg), iron (4.71–10.17 mg/kg), zinc (4.73–5.15 mg/kg), manganese (2.39–5.07mg/kg) and copper (1.07–3.59 mg/kg).

Results of the one-way pooled ANOVA showed a significant difference in the concentration means of all minerals (*p* ≤ 0.001) for the different date varieties except for zinc (*p* = 0.05). A post-hoc analysis indicated that boron, copper, iron, potassium, magnesium, manganese, and sodium in the Abu-Maan date were significantly lower than in most other varieties. 

### 2.2. Effect of Salinity Stress on the Mineral Composition and Percent Daily Values (%DV)

The %DV and mg/serving of each mineral in 13 date varieties at different salinities, and the mean mineral composition of date palm fruits cultivated at three salinity levels, 5, 10, and 15 dS m^−1^, are presented in Table 2 and Table 3.

Potassium (K)

Potassium was prevalent in considerable amounts in all 13 date varieties. The %DV of potassium ranged from 17% to 22%. Only the Nabtat-Saif variety was significantly affected by salinity. There was a significant drop in the potassium concentration from 7981 mg/kg at salinity level 5 dS m^−1^ to 6106 mg/kg at 10 dS m^−1^, then a significant increase to 7308 mg/kg at 15 dS m^−1^.

Phosphorus (P)

The %DV values of phosphorus ranged from 6% to 9%. Phosphorus levels decreased significantly only in the Khisab variety as the salinity increased from 5 to 10 dS m^−1^. 

Calcium (Ca)

The contribution of all 13 date varieties to the calcium daily intake was calculated to be relatively low, with the %DV ranging from 3.77% to 5.06%. There was a significant decrease in calcium levels in the Makhtoumi variety as salinity increased from 10 to 15 dS m^−1^, with no significant differences in calcium levels at salinity levels of 5 and 10 dS m^−1^.

Magnesium (Mg)

Palm dates had considerable amounts of magnesium in all 13 varieties (%DV, 10% to 14.5%) and none of the varieties showed a significant variation in the magnesium content with increasing salinity levels. 

Sodium (Na)

Dates are naturally low in sodium. Even after the palm trees were irrigated with salty water, the sodium levels were very low with a %DV contribution of 1%–2%. The level of salinity had no significant effect on the sodium content across all date varieties.

Boron (B)

All 13 date varieties were excellent sources of boron. The %DV ranged from 24.49% to 49.78%. The only significant effect of salinity on boron was observed in the Farad variety, where increased salinity levels from 5 to 15 dS m^−1^ significantly increased boron concentrations.

Iron (Fe)

The %DV of iron in dates varied from 3.06% to 6.61%. These levels do not contribute significantly to the daily intake. Salinity was not shown to have a significant effect on the content of iron across all the date varieties.

Zinc (Zn)

All 13 varieties showed low levels of zinc content with %DV contributions from 3.68% to 4%. Zinc was not significantly altered by an increased salinity across all the varieties. 

Manganese (Mn)

The %DV ranged from 8.52% to 18.09%. The salinity significantly affected manganese levels in the Khnizi and Farad varieties only. A significant decrease in manganese in the Khnizi (17.1% at 5 dS m^−1^ versus 10.6% at 10 dS m^−1^) and Farad (11.1% at 5 dS m^−1^ to 6.4% at 10 dS m^−1^) varieties was found when the salinity was increased from 5 to 10 dS m^−1^. However, at a salinity level of 15 dS m^−1^, the manganese level increased significantly in the Farad variety.

Copper (Cu)

Most of the date varieties were good sources of copper with a %DV ranging between 8.55% and 28.75%. The copper content of two of the 13 varieties (Khnizi and Makhtoumi) were significantly altered by salinity. The concentration of copper in the Khnizi variety decreased significantly with increased salinity levels (%DV of 28.64%, 10.4%, 7.44% at salinity levels 5, 10 and 15 dS m^−1^, respectively). In the Makhtoumi variety, the copper levels were significantly different at the three salinity levels, with the highest value at a salinity of 5 dS m^−1^ (33.06% DV) and the lowest at a salinity level of 10 dS m^−1^ (12% DV).

### 2.3. Grouping of Date Palm Varieties According to Their Fruit Quality under the Different Salinity Levels

In order to explain the phenotypic expression and adaptations to the different water irrigation salinity experienced, a genotype–environment interaction matrix was generated (Figure 1). The purpose of the matrix is to identify the adaptation of a variety to particular saline environments. Each tested variety’s average mineral value was compared with the average value measured for all the varieties in each environment. 

The adaptation of a variety i to the environment j (in terms of tolerance to salinity) can be evaluated by the sum of the terms of the variety (Gi) and the IGE [20]. The genotypes’ performance is detected through the interaction matrix by the circles’ variation in size and color. A black color corresponds to the case where the interaction between the variety and environment is positive, i.e., this variety’s fruit mineral content is above the average of all the genotypes in this environment. However, if the value of the variety’s mineral content is lower than the average of all the varieties, a green color is indicated. On the other hand, the circle’s diameter is even larger when the interaction is strong, i.e., the genotype is expressed better in this environment.

The interactions’ typology suggests that the tested genotypes show a specific adaptation. Subsequently, no genotypes showed a general adaptation to all the saline environments studied (Figure 1). Indeed, with each salinity level, some genotypes can grow and give a good fruit quality compared to all the genotypes. 

## 3. Discussion

Dates are an integral component of the Emirati daily meal plan and make a crucial contribution to the population’s nutritional intake. Dates are consumed as snacks, or as an ingredient in savory and dessert dishes. Dates are an important source of sugars, mainly the monosaccharides fructose and glucose, and the disaccharide sucrose. Moreover, dates are a rich fiber source, mostly insoluble, with small amounts of protein and fats [21]. In addition, dates are a rich source of a variety of vitamins and minerals, mostly vitamin B complex, vitamin C, selenium, copper, potassium and magnesium [19].

The impact of the irrigation of date palms with saline water on fruit quality, mainly in terms of the mineral content, is a very important indicator for their quality and our understanding of the physiological and biochemical processes involved under saline conditions. The results obtained showed that the mineral content in the varieties evaluated under all salinity levels was within the ranges reported by several other studies [22,23,24]. The 13 varieties of date palms exhibited diversity in their fruit mineral content. Significant variations for only a limited number of minerals were observed due to different varietal responses and the effects of salinity.

Overall, significant variations across the varieties were observed for most minerals. However, the impact of salinity was not similar for these varieties. A total of eight of the investigated varieties, mainly Ajwat AlMadinah, Naghal, Barhi, Shagri, Abu-Maan, Jabri, Sukkari, and Rothanwere were not affected by increased salinities up to 15 dS m^−1^. 

An increase in the salinity level resulted in slight changes in some minerals, but these were mostly not significant. The concentrations of iron, zinc, magnesium, and sodium remained unchanged in all date varieties as salinity levels increased. On the other hand, salinity stress did influence certain mineral compositions in specific varieties. Significant changes were observed in the boron concentration in the Farad variety, calcium in the Makhtoumi variety, copper in the Khnizi and Makhtoumi varieties, potassium in the Nabtat-Saif variety, manganese in the Khnizi and Farad varieties and phosphorus in the Khisab variety. The plant’s response to sodium is one of the critical influences of salinity. Results showed that most varieties have a low sodium concentration even at high salinity levels except for the Sukkari, Naghal and Barhi varieties. This indicates the latter varieties are not capable of excluding sodium. 

The fruit mineral composition varies within the same cultivated variety and partly responds to genetic effects. However, the performance may also vary depending on the environment. In addition, the variation in date palm minerals is largely due to the effects of abiotic constraints. As a result, some varieties display a high performance with some salinity levels, and a poor performance in others and the rankings between varieties are sometimes changed. This variability in the response of genotypes to salinity corresponds to the genotype–environment interaction.

The performance of all genotypes is highly variable as detected through the interaction matrix. This is the origin of the genotype x environment interaction (GEI), which is further confirmed by the reversal of classification for most genotypes according to the environment (qualitative interaction). This interaction, which induces a variable performance depending on the environment, is attributed to the differences in sensitivity levels to the irrigation water’s salinity vs. the plant’s defense mechanism. In arid environments such as in the UAE, the genotype x environment interaction is high; therefore, it is impossible to attribute the variation in mineral content between date palm varieties to the single effect of variety or salinity. Hence, it is important to take into account the adaptive characteristics which produce stable production in variable environments, i.e., large adaptability, or stable genotypic expression in a specific environment, i.e., specific adaptation [25,26]. The search for the genetic potential of mineral content in produced food must be accompanied at the same time by the search for performance stability and stress tolerance in the presence of a high GEI [27,28,29]. Moreover, the analysis of the behavior of genotypes according to the characteristics of the environment has long been a priority research topic.

The fluctuation in results from the phenotypic expression of tolerance to salinity through a complex set of biochemical and morpho-physiological properties is attributed to multiple mechanisms, including Na^+^ exclusion, Na^+^ sequestration in vacuoles, K^+^ retention, osmotic adjustment, and xylem control. The general sodium and potassium content, according to the three irrigation water salinity levels, are reversed. The average potassium content is higher in a salinity of 15 dS m^−1^ and the average sodium content is low under the same salinity level and vice versa.

In fact, the tolerant varieties try to limit Na^+^ and Cl^−^ while maintaining the absorption of nutrients such as K^+^, NO_3_^−^, and Ca^2+^ [15,30]. The mineral concentration in the fruits can be maintained under 10 dS m^−1^ and then it decreases or increases depending on the concentration in the soil root zone and the plant’s ability to take up minerals under a specific salt content in the root zone. Several regulatory mechanisms, based on the presence of calcium and potassium, and their role in stress signaling, such as that of Ca^2+^, have been identified as salt tolerance indicators [31]. Salinity tolerance was correlated with sodium-calcium or sodium-potassium selectivity based on a simple exchange of ions on the plasma membrane’s surface [32,33]. Therefore, the Na^+^/K^+^ pump works very well under 10 dS m^−1^. Consequently, the concentration of an element becomes higher as salinity increases to 5 dS m^−1^. However, above 10 dS m^−1^, this tolerance mechanism can no longer work; consequently, the concentration of particular beneficial elements for plants will be reduced. This nutritional stress becomes one of the significant effects of salinity after osmotic stress. Consequently, a specific mineral can increase when salinity increases from 5 to 10 dS m^−1^ and this is probably due to a tolerance mechanism such as potassium retention; then the specific mineral decreases when salinity increases to 15 dS m^−1^. This indicates that the salinity tolerance threshold is 10 dS m^−1^ for this specific genotype. However, for other varieties, we may observe a decrease as salinity increases from 5 to 10 dS m^−1^, indicating a tolerance threshold of 5 dS m^−1^ due to inactivation of the potassium retention mechanism.

This study of the long-term effect of saline water irrigation on date palm fruit quality highlighted the instability of Jabri, Fard, Khisab and Nabtat-Saif varieties in terms of their mineral content. Meanwhile, Maktoumi, Barhi Ajwat Al Madinah, Khinizi and Shagri varieties showed fewer interactive behaviors with the salinity variation, and their mineral content was similar to the general mean. Thus, varietal experimentation and varietal performance analysis is an approach that has been widely used for breeding and selection with noticeable results [34]. It involves the establishment of trials as the main tool of research. Experiments are based on varietal trials (grouping several genotypes or varieties), multi-local, very general multi-year and multi-treatment trials, to evaluate the performance of different genotypes.

The %DV, among all the date varieties, remained within the same category (low, good or high) despite some observed changes in the %DV with increased salinity. The only changes in %DV categories were observed for copper in the Khnizi and Makhtoumi varieties (high to good), and for managenese in the Farad variety (good to low), as salinity increased from 5 to 10 dS m^−1^. Calcium, iron, sodium, and zinc showed a low %DV. Phosphorus was marginally a good source; magnesium and manganese recorded a good contribution to the dietary intake. Boron, copper, and potassium showed a high %DV across the different date varieties.

## 4. Materials and Methods

### 4.1. Experimental Setting

A long-term experiment using local and imported date palm varieties was conducted in 2001 at the ICBA experimental station (25 13″ N and 55 17″ E) known to be one of the harshest environments in the region [11]. Eighteen local and imported date palm varieties were grown under three treatments differing by the level of salinity in the irrigation water (5, 10, 15 dS m^−1^) with five repetitions (five trees per treatment). Out of the eighteen varieties, thirteen were the subject of this study (Table 4).

The experiment was conducted using a split plot design. The trial field was divided into three subplots. Each subplot was subjected to one water salinity treatment. The planting arrangement was systematic, planting in rows within each salinity level, with a tree planting spacing of 8m by 8 m. In addition, a gap of 20 m was kept between each plot group of five plants. Three salinity levels (5, 10, 15 dS m^−1^) of irrigation water were applied to each plot. Irrigation treatments were arranged in a randomized complete block design with 5 replicates per treatment. Each treatment plot consisted of 3 subplots, each containing 5 trees. The trial site soil is Carbonatic, Hyperthermic Typic Torripasmment, having a negligible level of inherent soil salinity (0.2 dS m^−1^). In addition, soil samples were collected at 0–60 cm to monitor the root zone salinity as a result of irrigation with saline water. As expected, the highest salinity levels were found, especially after the trial period, in the plot where highly saline water (ECe 15 dS m^−1^) was applied (Table 5).

Organic compost manure was applied at the rate of 20 kg per tree per year during the last two weeks of October and NPK fertilizer was applied early in October and December yearly at the recommended level as per normal agronomical practices in the UAE. Pollination extended from early February to late March and the harvest occurred generally during the summer season (July–August). Trees were irrigated using a bubble system twice a day for 20 min each. For the irrigation, weather data was collected from a weather station located at ICBA (LiCor 1200, LiCor Inc., Lincoln, NE 68504-5000, USA) and was used to estimate the reference evapotranspiration (ET) according to the Penman–Monteith evapotranspiration FAO-56 method, and then the total water supplied was determined for each month to obtain the date palm water requirement (Table 6). Irrigation was applied using a bubbler system.

### 4.2. Salinity Treatments

Three salinity treatments were established of 5, 10 and 15 dS m^−1^. The three levels represent the levels expected to achieve a substantial yield reduction, and to meet 50% yield reduction thresholds. The 5, 10 and 15 dS m^−1^ irrigation salinity was accomplished by mixing highly saline groundwater (with an ECw up to 25 dS m^−1^, SAR > 26 mmol/L with Na^+^ and Cl^−^ concentrations higher than 190 meq/L and pH = 7.6) with low salinity municipal water of less than 2 dS m^−1^, which alone, was the lowest salinity water available (SAR = 4 mmol/L with Na^+^ and Cl^−^ concentrations lower than 11 meq/L and pH = 8.5). The three salinity levels were constantly maintained throughout the cropping season during all the years.

### 4.3. Mineral Analysis

Date samples were collected in the two growing seasons, 2016 and 2017 after harvest at the “Tamar” stage. A total of 117 samples, consisting of three replicates of the 13 different varieties (5 from the UAE, 7 from the Kingdom of Saudi Arabia (KSA) and 1 from Iraq) grown at the three salinity levels were analyzed. All samples were washed with deionized water and dried at 100 °C for 24 h until they attained a constant mass. Each sample was then powdered, sieved and stored in a plastic bag for metal analysis.

All glassware and digestion vessels were soaked in 20% nitric acid and rinsed with ultrapure water (Millipore Elix Advantage Water Purification System, Millipore, MA, USA). Multi-element standard solutions were prepared by diluting 1000 mg/L stock solutions (Fluka traceCert Ultra, Sigma-Aldrich) with a 5% HNO_3_ solution (trace metal concentrated, supra pure Merk).

About 0.5 g of each sample was accurately weighed into a digestion vessel (MARSXpress), followed by the addition of 5 mL of nitric acid (HNO_3_) (trace metal concentrated, supra pure Merk) and 2 mL of 30% hydrogen peroxide (H_2_O_2_) (Sigma-Aldrich). The mixture was subjected to microwave-assisted digestion in a MARS microwave digestion system (CEM Corporation Matthews, USA) at 200 °C and 70 Bar for 55 min. At the end of the digestion program, the samples were filtered and quantitatively transferred to 50 mL volumetric flasks and diluted with water. The concentration of the minerals in the sample was determined using Inductively coupled plasma—optical emission spectrometry (ICP-OES) (Model no. 700 series, Agilent Technology, Santa Clara, CA, USA).

All quality control and assurance measures were taken, including calibration check measures, determination of the method’s limit of quantification (MLQ), and replicating sample analyses. The concentration of the minerals is expressed as the mean value (mg/kg of dry weight) ± SD of replicates of the same date variety, collected from the same row at the same salinity level.

### 4.4. Calculation of Percent Daily Values (%DV)

Each date weighs on average 9 g, as per FAO. The average consumption of dates in the UAE is 8 dates per day (72 g) (survey reference in the UAE). All calculations are based on an average daily consumption of 72 g. One serving of dates is 3 dates (27 g). To find the %DV of a nutrient, the amount of the nutrient in a serving size is divided by the daily value from the Dietary Reference Intake (DRI) tables [35], then multiplied by 100. To identify the magnitude of the contribution of each of the minerals to the daily intake, %DV was calculated for all minerals. As per FDA, a %DV of a nutrient of 5% or less per serving is considered low, between 5% and 10% is a marginally good source, between 10% and 19% is good, while values of 20% or more is high [36].

### 4.5. Statistical Analysis

The collected data were coded, entered, and analyzed using the statistical package SPSS version 26. Statistical tests with *p*-values < 0.05 were considered statistically significant. The normality of the salinity for all minerals and multinutrients was checked using the Shapiro–Wilk test (*p*-value > 0.05). Kurtosis and skewness, histogram, and Q_Q plots were also used to check the normality of all variables. Moreover, data was cleaned of outliers, and the assumption of homogeneity of variances between the groups was checked using Levene’s Test. Therefore, one-way Analysis of Variance (pooled ANOVA) was carried out to test the equality of means across the different salinity levels. Duncan’s multiple range test at the 0.05 level was used to determine the statistical difference between the means (Appendix A Table 2 with Pooled SD).

To study the genotype x environment (salinity) interaction, the mineral composition (MC) of a variety i in an environment (salinity) j in a row k can be expressed as follows:MCijk = μ + Gi + Ej + ExBjk + GxEij + εijk
where μ is the mean genotype MC observed in the whole experiment, Gi is the mean effect of the genotype i, Ej is the mean effect of the environment (salinity) j, ExBjk is the effect of the row k in the environment (salinity) j, GxEij is the particular effect of the genotype i in the environment j, and εijk is the residue observed for the genotype i in the row k of the environment j.

Each date palm variety (i) was described by its mean MC under all salinity (μ + Gi) conditions. Then, the adaptation of the variety i to an environment j (in terms of mineral composition) was assessed by the sum of the genotypic (Gi) and the interaction (GxEij) terms [20]. According to their adaptations to the different environments of the multiple irrigation water salinity levels, the varieties were grouped by a hierarchical ascending cluster analysis (HCA) based on Euclidean distances between varieties and Ward’s method of grouping minimum variance [37]. We created a Genotype x Environment (Salinity) Interaction (GEI) Matrix with the HCA of the fruit mineral content expression of the 13 date palm varieties. The hclust function (library Hmisc and FactoMineR) and plot visualization package (FunVisuModIGE.r), with the statistical software R version 4.0.2. was used.

## 5. Conclusions

Screening date palm varieties for their salinity adaptive capacity showed that certain varieties, mainly Ajwat AlMadinah, Naghal, Barhi, Shagri, Abu Maan, Jabri, Sukkari and Rothan, can endure a relatively high soil salinity level with no visible effect on the mineral content. Results also suggest that all examined varieties remain good sources of dietary potassium, magnesium, manganese and boron even at high salinity levels.

It was evident that no genotypes showed a general adaptation to all the saline environments studied. However, Barhi, Ajwat Al Madinah, Khinizi, Maktoumi and Shagri varieties were more stable and showed fewer interactive behaviors with the salinity variation. In arid environments such as in the UAE, the genotype x saline-environment interaction was found to be high, which makes it impossible to attribute the variation in mineral content to a single varietal or salinity effect. This trend results from the negative salinity effect and the counter-effort by the phenotypic expression of tolerance to salinity through a complex set of biochemical and morpho-physiological properties attributed to multiple mechanisms, including Na^+^ exclusion, K^+^ retention, and osmotic adjustment.

Overall, results obtained from this research provide a comprehensive view of salinity tolerance in date palms. Screening date varieties for their salt tolerance can yield valuable information on their adaptive mechanisms and their interaction with other nutrients. It provides resources for improved date palm production as an alternative halophyte crop. This in turn will allow for better utilization of seawater irrigation in marginal areas. In addition, evaluating the effect of salinity stress on the mineral composition of date palm fruits grown under high salinity conditions is crucial for understanding the health risks and nutritional benefits of this important crop. Finally, research should be directed towards understanding the nature of the salt-adaptation mechanism in order to develop future date palm varieties that can tolerate excessive soil salinity.

## Figures and Tables

**Figure 1 molecules-26-07361-f001:**
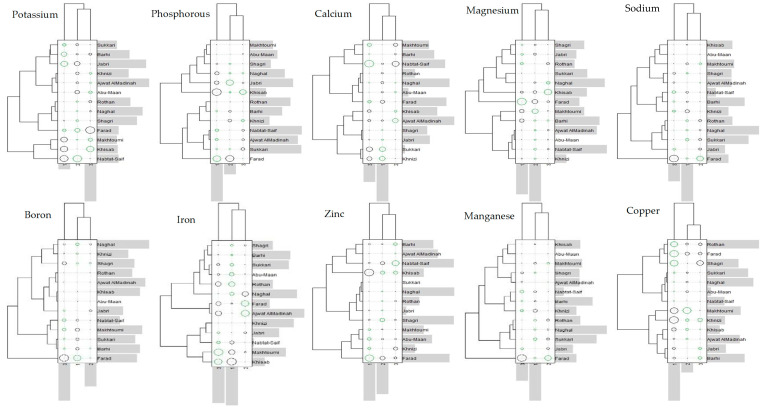
Genotype x Environment (Salinity) Interaction (GEI) Matrix of the fruit mineral content expression of 13 date palm varieties grown using three irrigation water salinity levels corresponding to electrical conductivities of 5, 10 and 15 dS m^−1^ denoted as 1, 2, and 3, respectively.

**Table 1 molecules-26-07361-t001:** The mean mineral composition of date palm fruits grown under different salinity conditions.

Date Type (Origin)	Boron mg/kg	Calcium mg/kg	Copper mg/kg	Iron mg/kg	Potassium mg/kg	Magnesium mg/kg	Manganese mg/kg	Sodium mg/kg	Phosphorus mg/kg	Zinc mg/kg
Ajwat AlMadinah (KSA)	12.08 ± 2.19	766.00 ± 164.27	2.66 ± 1.47	10.08 ± 3.23	8293.69 ± 1072.52	674.63 ± 70.31	3.67 ± 1.08	330.91 ± 115.92	836.65 ± 107.07	5.02 ± 0.37
Naghal (UAE)	12.84 ± 3.22	745.57 ± 162.43	3.28 ± 1.17	10.17 ± 2.61	8049.67 ± 947.30	709.16 ± 124.26	5.07 ± 1.27	429.98 ± 169.45	784.51 ± 131.33	5.14 ± 0.48
Khnizi (UAE)	10.22 ± 3.21	571.95 ± 140.64	2.04 ± 1.44	9.60 ± 2.65	7470.83 ± 689.52	547.40 ± 93.34	3.50 ± 1.39	296.17 ± 122.06	706.56 ± 125.49	4.83 ± 0.39
Barhi (Iraq)	11.50 ± 3.02	692.34 ± 139.36	2.85 ± 1.23	9.25 ± 1.40	7545.97 ± 966.20	686.89 ± 110.67	4.31 ± 1.35	407.81 ± 73.49	765.41 ± 110.13	4.98 ± 0.42
Makhtoumi (KSA)	11.71 ± 3.13	675.06 ± 90.27	2.71 ± 1.31	8.31 ± 2.96	7052.90 ± 885.27	617.55 ± 109.06	3.78 ± 1.59	352.56 ± 118.44	725.36 ± 127.75	4.90 ± 0.30
Farad (UAE)	11.23 ± 5.62	736.45 ± 218.20	1.07 ± 0.94	7.50 ± 6.08	7426.46 ± 1241.74	593.33 ± 179.85	4.11 ± 1.78	340.04 ± 107.18	611.60 ± 96.66	5.07 ± 0.38
Khisab (UAE)	11.48 ± 2.09	704.55 ± 114.16	2.10 ± 1.15	6.17 ± 3.11	7106.56 ± 859.63	632.70 ± 146.38	3.69 ± 1.23	427.47 ± 91.03	694.27 ± 146.64	4.85 ± 0.45
Nabtat-Saif (KSA)	9.07 ± 2.23	734.64 ± 137.47	1.88 ± 0.34	4.92 ± 2.43	7131.74 ± 1011.44	717.26 ± 84.26	2.88 ± 1.12	362.83 ± 107.62	852.03 ± 84.75	5.03 ± 0.41
Shagri (KSA)	9.89 ± 4.12	662.20 ± 146.89	2.47 ± 1.10	7.08 ± 2.01	7756.75 ± 854.80	625.65 ± 62.47	3.50 ± 1.29	345.00 ± 92.18	712.42 ± 131.47	5.15 ± 0.37
Abu-Maan (KSA)	6.32 ± 2.06	595.34 ± 114.75	1.24 ± 0.72	4.71 ± 2.89	6306.95 ± 1116.16	496.55 ± 65.09	2.39 ± 1.03	311.13 ± 82.50	744.15 ± 133.03	4.96 ± 0.34
Jabri (UAE)	9.83 ± 2.28	679.48 ± 163.58	2.00 ± 1.00	7.97 ± 0.43	8187.88 ± 991.21	587.77 ± 66.91	3.34 ± 0.87	245.44 ± 42.56	801.63 ± 151.29	4.77 ± 0.31
Sukkari (KSA)	10.88 ± 2.39	581.47 ± 130.79	3.06 ± 0.84	9.09 ± 2.99	7064.43 ± 1456.03	635.54 ± 104.72	4.54 ± 1.55	207.29 ± 57.08	847.53 ± 110.22	4.73 ± 0.25
Rothan (KSA)	10.55 ± 2.62	620.15 ± 221.21	3.59 ± 0.93	7.14 ± 2.33	7591.55 ± 907.08	580.71 ± 82.31	3.71 ± 0.94	230.25 ± 122.83	803.12 ± 181.15	4.89 ± 0.32
*p*-Value	<0.0001	0.001	<0.0001	<0.0001	<0.0001	<0.0001	<0.0001	<0.0001	0.001	0.05

**Table 2 molecules-26-07361-t002:** The mean concentration of minerals in each date palm variety at three salinity levels.

Date Type	Salinity	B mg/kg	Ca mg/kg	Cu mg/kg	Fe mg/kg	K mg/kg	Mg mg/kg	Mn mg/kg	Na mg/kg	P mg/kg	Zn mg/kg
Ajwat AlMadinah	1	11.1	835	3.5	11.7	8184	678	3.7	339	799	5.1
	2	12.3	652	2.2	8.3	8518	653	3.1	234	835	5.1
	3	12.9	820	2.3	12.9	8179	693	4.2	381	870	4.9
Naghal	1	9.5	831	3.6	9.1	7959	726	4.7	419	874	5.2
	2	13.8	693	3.5	11.0	7780	745	5.5	532	739	5.1
	3	14.4	728	2.9	9.8	8409	656	5.0	345	719	5.1
Khnizi	1	7.7	513	3.58 ^a,b^	10.1	7377	568	4.79 ^a,b^	301	723	4.7
	2	10.7	543	1.30	9.1	7827	527	2.96	319	772	4.8
	3	11.8	655	0.93	9.6	7327	541	2.68	272	636	5.0
Barhi	1	12.7	712	3.78	9.04	6882	752	4.99	400	744	5.13
	2	10.8	701	2.88	8.68	7753	661	4.33	459	806	5.09
	3	11.04	664	2.04	9.89	8003	648	3.73	372	746	4.71
Makhtoumi	1	13.6	707	4.2 ^a,b^	10.7	7732	685	4.75	442	759	4.85
	2	11.6	712 ^c^	1.5^c^	8.6	6806	549	2.93	320	705	4.96
	3	10.1	600	2.4	6.7	6662	630	3.79	317	714	4.88
Farad	1	5.9 ^b^	842	0.4	7.7	6696	495	3.1 ^a,b^	241	510	4.7
	2	10.4	695	1.5	4.2	6489	680	1.8	337	758	5.4
	3	17	680	1.2	8.5	8226	630	5.6	407	597	5.2
Khisab	1	11.3	719	2.8	8.9	7869	675	4.3	501	866 ^a,b^	5.4
	2	11.4	628	1.05	5.2	7057	692	3.3	424	653	4.7
	3	11.7	766	n.d.	4.5	6393	531	3.5	358	564	4.6
Nabtat-Saif	1	9.9	797	2.2	6.5	7981 ^a,b^	731	3.7	354	810	5.3
	2	10.0	810	1.8	4.1	6106 ^c^	685	3.0	286	860	5.4
	3	7.3	597	1.7	4.2	7308	735	1.9	449	886	4.7
Shagri	1	8.8	673	2.1	6.8	7963	601	3.4	376	774	5.3
	2	9.5	641	2.6	7.1	7435	652	3.3	341	675	4.9
	3	14.2	683	3.1	7.6	8095	614	4.3	265	697	5.2
Abu-Maan	1	6.3	666	1.55	3.8	6137	504	2.5	320	762	4.8
	2	6.1	525	1.47	4.5	6792	476	2.2	297	715	5.1
	3	6.6	594	0.69	5.8	5992	509	2.4	316	755	5.0
Jabri	1	8.2	677	2.9	8.3	7397	554	3.5	232	920	4.9
	2	11.4	653	1.8	8.0	8734	635	3.9	254	687	4.7
	3	8.8	699	1.2	7.4	8432	574	2.6	250	837	4.7
Sukkari	1	12.1	515	3.3	8.4	6636	639	3.8	233	828	4.8
	2	10.4	549	3.2	8.6	7359	649	5.1	188	819	4.7
	3	10.2	679	2.7	10.2	7198	618	4.7	201	906	4.7
Rothan	1	9.3	686	3.2	5.9	7454	545	3.2	177	805	4.9
	2	11.1	571	3.9	6.5	7245	586	4.3	263	820	4.8
	3	11.2	603	3.7	9.0	8075	610	3.6	251	784	4.9

Salinity level: 1 = 5 dS m^−1^; 2 = 10 dS m^−1^; 3 = 15 dS m^−1^. ^a^ Significantly different between salinity levels 1 and 2. ^b^ Significantly different between salinity levels 1 and 3. ^c^ Significantly different between salinity levels 2 and 3. n.d.: Not determined.

**Table 3 molecules-26-07361-t003:** Percent Daily Values (%DV) and mg/serving of each mineral in date varieties at different salinities.

	Boron		Calcium		Copper		Iron		Potassium		Magnesium		Manganese		Sodium		Phosphorus		Zinc	
	Per Serving/Day																			
Date Type	%DV	mg	%DV	mg	% DV	mg	% DV	mg	%DV	mg	%DV	mg	%DV	mg	%DV	mg	%DV	mg	%DV	mg
Ajwat AlMadinah	46.84	0.33	5.06	20.68	21.30	71.90	6.55	0.27	22.00	224.00	13.60	18.20	13.07	0.10	2.00	9.00	9.00	23.00	3.90	0.14
Naghal	49.78	0.35	4.92	20.13	26.27	88.66	6.61	0.27	21.00	217.00	14.30	19.10	18.09	0.14	2.00	12.00	8.00	21.00	3.99	0.14
Khnizi	39.61	0.28	3.77	15.44	16.34 *	55.14 *	6.24	0.26	20.00	202.00	11.00	14.80	12.49 *	0.09 *	1.00	8.00	7.00	19.00	3.76	0.13
Barhi	44.57	0.31	4.57	18.69	22.79	76.93	6.01	0.25	20.00	204.00	13.90	18.50	15.38	0.12	2.00	11.00	8.00	21.00	3.87	0.13
Makhtoumi	45.42	0.32	4.46 *	18.23 *	21.68 *	73.16 *	5.40	0.22	19.00	190.00	12.50	16.70	13.46	0.10	2.00	10.00	7.00	20.00	3.81	0.13
Farad	43.53 *	0.30 *	4.86	19.88	8.55	28.84	4.87	0.20	19.00	201.00	12.00	16.00	14.64*	0.11*	2.00	9.00	6.00	17.00	3.94	0.14
Khisab	44.50	0.31	4.65	19.02	16.80	56.70	4.01	0.17	19.00	192.00	12.80	17.10	13.15	0.10	2.00	12.00	7.00 *	19.00 *	3.77	0.13
Nabtat-Saif	35.16	0.24	4.85	19.84	15.03	50.71	3.20	0.13	19.00 *	193.00 *	14.50	19.40	10.27	0.08	2.00	10.00	9.00	23.00	3.91	0.14
Shagri	38.35	0.27	4.37	17.88	19.78	66.76	4.60	0.19	20.00	209.00	12.60	16.90	12.49	0.09	2.00	9.00	7.00	19.00	4.00	0.14
Abu-Maan	24.49	0.17	3.93	16.07	9.90	33.41	3.06	0.13	17.00	170.00	10.00	13.40	8.52	0.06	1.00	8.00	8.00	20.00	3.86	0.13
Jabri	38.12	0.27	4.48	18.35	15.98	53.94	5.18	0.22	21.00	221.00	11.90	15.90	11.92	0.09	1.00	7.00	8.00	22.00	3.71	0.13
Sukkari	42.20	0.29	3.84	15.70	24.47	82.60	5.91	0.25	19.00	191.00	12.80	17.20	16.18	0.12	1.00	6.00	9.00	23.00	3.68	0.13
Rothan	40.91	0.28	4.09	16.74	28.75	97.04	4.64	0.19	20.00	205.00	11.70	15.70	13.24	0.10	1.00	6.00	8.00	22.00	3.80	0.13

* Significantly different values at salinity levels of 5, 10, and 15 dS m^−1^, respectively. The average is represented in the table. The content in mg/serving/day across salinity levels of 5, 10 and 15 dS m^−1^, respectively: boron in Farhad (0.16, 0.28, 0.46); calcium in Makhtoumi (19.09, 19.22, 16.20); copper in Khnizi (96.66, 35.10, 25.11); copper in Makhtoumi (113.40, 40.50, 64.80); potassium in Nabtat-Saif (215.00, 165.00, 197.00); manganese in Khnizi (0.13, 0.08, 0.07); manganese in Farad (0.08, 0.05, 0.15); phosphorus in Khisab (23.00, 18.00, 15.00). The %DV per serving/day across salinity levels of 5, 10 and 15 dS m^−1^, respectively: boron in Farhad (22.90, 40.30, 65.90); calcium in Makhtoumi (4.67, 4.7, 3.96); copper in Khnizi (28.64, 10.40, 7.44); copper in Makhtoumi (33.60, 12.00, 19.20); potassium in Nabtat-Saif (21.00, 16.00, 19.00); manganese in Khnizi (17.10, 10.60, 9.60); manganese in Farad (11.10, 6.40, 20.00); phosphorus in Khisab (9.00, 7.00, 6.00).

**Table 4 molecules-26-07361-t004:** Date palm varieties used in the experiment and their origin, approximate potential yield, maturity, and distribution in the UAE.

Date Type	Origin	Approximate Yield Potential kg/Tree	Maturity Group (Early, Mid, Late)	Distribution in the UAE
Ajwat AlMadinah	KSA	60–70	Mid	Very limited
Naghal	UAE	40–60	Very early	All UAE
Khnizi	UAE	60–70	Mid to late	All UAE
Barhi	Iraq	80/120	Mid to late	All UAE
Makhtoumi	KSA	40–60	Mid	In some region
Farad	UAE	70–90	Mid to late	All UAE
Khisab	UAE	100–120	Very late	Very limited
Nabtat-Saif	KSA	35–60	Mid	In some region
Shagri	KSA	50–60	Mid	-
Abu-Maan	KSA	50–70	Mid	In some region
Jabri	UAE	40–60	Late	All UAE
Sukkari	KSA	50–70	Mid	Very limited
Rothan	KSA	60–70	Mid	Very limited

Source: Date palm varieties in the United Arab Emirates, ministry of agriculture and fisheries, UAE.

**Table 5 molecules-26-07361-t005:** Soil properties before and after the growing season 2016–2017.

Period	Salinity	Clay %	Silt %	Sand %	pH	ECe (dS m^−1^)
Soil testing of the experiment before the two growing seasons, 2016 and 2017 *	1	0.55	0.67	98.78	7.33	0.66
	2					1.00
	3					1.33
Soil testing of the experiment after the two growing seasons, 2016 and 2017, during the 2018 season.	1	0.50	0.90	98.60	7.47	0.994 ± 0.09
	2	0.69	0.36	98.95	7.38	1.02 ± 0.2
	3	0.27	0.76	98.97	7.38	3.24 ± 0.5

* Dates should be considered as indicative values. Salinity level: 1 = 5 dS m^−1^; 2 = 10 dS m^−1^; 3 = 15 dS m^−1.^

**Table 6 molecules-26-07361-t006:** Irrigation scheduling during the two growing seasons, 2016 and 2017.

Month	January	February	March	April	May	June	July	August	September	October	November	December
Liters of water per day per tree	132	170	216	251	276	285	278	254	219	177	143	122

## Data Availability

The data presented in this study are available on request from the corresponding author.

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
