# Peer review of "The Mineral Composition of Date Palm Fruits (Phoenix dactylifera L.) under Low to High Salinity Irrigation"

_molecules, 2021, doi:10.3390/molecules26237361_

Round 1

Reviewer 1 Report

I think it is a complete and potentially interesting study for the readers of the magazine. I think it is relevant to study water salinity and how this characteristic affects different crops.
I consider that the article is optimal for publication in the journal, but I would like a more concrete conclusion where they define concrete ideas that they have found when analyzing the data. I consider that the suitability of knowing the organoleptic behavior of any food in relation to the characteristics of water is very relevant, as they argue in their conclusion, but I would like a small paragraph where they explain concrete conclusions of their study.

Author Response

Dear Reviewer

We would like to thank you for your feedback, and we appreciate the insightful comments on improvements to our paper. We have changed the conclusion and the abstract to provide more support to the findings and clearer response to the research objectives. Those changes were made in track-changes in the resubmitted manuscript. English language and spell were checked for errors.

Best Regards

Rania Dghaim (on behalf of the authors)

Reviewer 2 Report

This is an interesting case study which is highly significant for scientists involved in date palm improvement programme and date growers. However, some changes, revisions, clarifications etc. can enhance the quality of this article. Therefore, this manuscript can be considered for publication in biomolecules after minor revisions.

Specific comments

1.To unravel the interactive association between the minerals, association analysis would be applied and mentioned in graphical of tabular form.

2.Group comparison (between local and imported varieties) must be performed to clarify the differences between minerals in local and imported varieties

3.Pooled ANOVA of the experimental design would be clearly mentioned in the result section to see the treatments effect on minerals   

4.Explain why you are not using mixed models? How you control pseudo replication in your model?

5.All the tables must be in three line format.

6.Quality of Figures has to be improved. Remember that everything written inside the figures must be clearly readable and understandable by the readers.

7.Conclusion is not providing any take home message. The conclusions section should not be a summary of your study. The conclusions section should illustrate the mechanistic links of findings obtained under applied treatments. In such type of studies some important recommendations would be emerged and listed point wise, for example suitability of date varieties according to available saline irrigation water (5EC, 10 EC, and 15 EC) considering the rich minerals i.e. copper, potassium and magnesium, which could be helpful for date growers.

General comments

Lines 12-24: Improve the abstract, it should indicate the most important positive results.

Line 43: Abbreviations and acronyms must be explained and placed in parentheses only the first time they are used (ex. MENA)

Lines 44-47: Statements must be rephrased as both the statements will create controversy.

Line 59: Reworded as overexploitation of groundwater resources.

Lines 75-79: Rephrased the paragraph meaningfully.

Lines 90-92: Hypothesis of taking up research must be clearly defined

Line 103: DMRT test must be applied to differentiate the mean differences in table 1

Line 105: Significance is not reflected in the table 1, please justify the statement

Lines 111-112: Where the results of one-way ANOVA are mentioned and how did you applied one-way ANOVA for such type of factorial experiments, please justify.

Lines 191-194: Circle color must be rewritten in reversed order for good combination

Line 303: In other sections of the manuscript 13 varieties are mentioned, please take into consideration. It would be better if available passport data of the varieties (developmental year, pedigree, local /imported, yield potential/ha, maturity group-early, mid, late, targeted region etc.) is given in tabular form.

Lines 305-306: Please clearly mentioned in which design experiment was conducted i.e. strip plot design or split plot design.

Lines 312-313: It will be better to present the soil properties (before and after the experiment) in tabular form and also clearly mentioned the long term effect of saline water in build-up of soil salinity in the experiment.

 Line8 318-319 : Water applied must be mentioned in quantifiable  term (l/tree).

Lines 330-331: sampling stage of the fruit would be clearly mentioned. According to your experiment total number of samples would be 13V x5R x3S = 195 per year, please mentioned how sampling was done??

Lines 367-369: Please mentioned how many traits were following normal distribution and which transformation method was applied to traits which do not follow the normality?? 

Line 389: Please mention the name of the particular package. 

Author Response

Dear Reviewer

We would like to thank you for the insightful and constructive comments. The manuscript is substantially improved as a result. We have changed the conclusion and abstract to provide more support to the findings and clearer response to the research objectives. Those changes were made in track-changes in the resubmitted manuscript. English language and spell were checked for errors. Kindly find below authors’ response to the reviewer general and specific comments.

Please note that line numbers used in the response correspond to numbers in the manuscript with (No Markup).

Best Regards

Rania Dghaim 
